# KnapSpec: Self-Speculative Decoding
# via Adaptive Layer Selection as a Knapsack Problem

Seongjin Cha [1] [*]  Gyuwan Kim [2] [*]  Dongsu Han [1]  Tao Yang [2]  Insu Han [1]

## Abstract

Self-speculative decoding (SSD) accelerates LLM inference by skipping layers to create an efficient draft model, yet existing methods often rely on static heuristics that ignore the dynamic computational overhead of attention in long-context scenarios. We propose KnapSpec, a training-free framework that reformulates draft model selection as a knapsack problem to maximize tokens-per-time throughput. By decoupling Attention and MLP layers and modeling their hardware-specific latencies as functions of context length, KnapSpec adaptively identifies optimal draft configurations on the fly via a parallel dynamic programming algorithm. Furthermore, we provide the first rigorous theoretical analysis establishing cosine similarity between hidden states as a mathematically sound proxy for the token acceptance rate. This foundation allows our method to maintain high drafting faithfulness while navigating the shifting bottlenecks of real-world hardware. Our experiments on Qwen3 and Llama3 demonstrate that KnapSpec consistently outperforms state-of-the-art SSD baselines, achieving up to $1.47\times$ wall-clock speedup across various benchmarks. Our plug-and-play approach ensures high-speed inference for long sequences without requiring additional training or compromising the target model's output distribution. Our official implementation is available at https://github.com/kaist-flexml-lab/knapspec.

[*]Equal contribution [1]School of Electrical Engineering, KAIST, Daejeon, South Korea [2]Department of Computer Science, University of California, Santa Barbara, CA, USA. Correspondence to: Insu Han <insu.han@kaist.ac.kr>.

## 1. Introduction

Large language models (LLMs) have become central components in modern AI systems, powering applications well beyond traditional natural language processing, including code generation (Dong et al., 2025; Joel et al., 2024), multimodal reasoning (Lu et al., 2022), and interactive agents (Park et al., 2023). As model sizes and context lengths continue to grow, inference cost has emerged as a major bottleneck for scalable deployment. During the generation stage, producing each new token typically requires substantial computation thereby increasing inference latency and computational cost. Efficient decoding strategies are therefore essential for enabling practical use of LLMs.

Speculative Decoding (SD) (Leviathan et al., 2023; Chen et al., 2023; Xia et al., 2023; Choi et al., 2025) has recently gained attention as an effective strategy for accelerating sequence generation without modifying model parameters. Typically, a lightweight draft model generates tokens sequentially in an autoregressive manner, and a target model then verifies multiple proposed tokens in parallel. This enables substantial speedups by amortizing the expensive verification computation across multiple tokens. Concurrently, recent frameworks (Sun et al., 2024; Sadhukhan et al., 2025; Cha et al., 2026) adapt this paradigm to long contexts by optimizing KV cache management and attention operations. In most existing speculative decoding approaches, the draft and target models are distinct, with the draft model chosen to be significantly cheaper than the target model. However, relying on two distinct models introduces practical and algorithmic challenges: maintaining and deploying an additional draft model increases system complexity, and distribution mismatch between the draft and target models can significantly reduce acceptance rates.

Self-Speculative Decoding (SSD) (Elhoushi et al., 2024; Liu et al., 2024; Xia et al., 2025; Chen et al., 2025; Zhang et al., 2024) mitigates these limitations by using a single model for drafting and verification. A partial computation of the target model can already produce token predictions that closely approximate the full model. By sharing parameters and representations, SSD preserves distributional similarity and removes the need for an auxiliary model. This approach avoids the high cost of training or distilling

*Table 1.* Comparison of KnapSpec with existing self-speculative decoding methods. KnapSpec is the first to offer a training-free, module-level separable search that adaptively reconfigures the draft model based on context length.

| Methods | Training-Free | Attn/MLP Separable | Adaptive Search | Context-Length Aware | Search Strategy |
|---|---|---|---|---|---|
| LayerSkip (Elhoushi et al., 2024) | ✗ | ✗ | ✗ | ✗ | Static Early-Exit |
| Kangaroo (Liu et al., 2024) | ✗ | ✗ | ✗ | ✗ | Fixed Sub-network |
| DEL (Zarch et al., 2025) | ✗ | ✗ | ✓ | ✗ | Dynamic Early-Exit |
| ASD (Metel et al., 2024) | ✓ | ✓ | ✓ | ✗ | Rule-based Pruning |
| Draft & Verify (Zhang et al., 2024) | ✓ | ✓ | ✗ | ✗ | Bayesian Optimization |
| SWIFT (Xia et al., 2025) | ✓ | ✓ | ✗ | ✗ | Bayesian Optimization + Random Search |
| CLaSp (Chen et al., 2025) | ✓ | ✗ | ✓ | ✗ | Dynamic Programming |
| **KnapSpec (Ours)** | ✓ | ✓ | ✓ | ✓ | **Knapsack Optimization** |

separate draft models. Furthermore, SSD approaches that select a sub-network of the target model can significantly reduce verification overhead by reusing intermediate states, effectively streamlining the inference pipeline.

## 1.1. Related Work

LayerSkip (Elhoushi et al., 2024) proposes a training-based SSD approach that enables early-exit inference by applying layer dropout and a shared early-exit loss for fine-tuning, allowing the model to exit at early layers for drafting. Similarly, Kangaroo (Liu et al., 2024) adopts a double early-exit strategy that utilizes a shallow sub-network of the target LLM; it requires a fine-tuned adapter module to bridge the gap between the sub-network and the full model. In contrast, several recent works focus on training-free, plug-and-play solutions that require no additional parameters or auxiliary training. Draft&Verify (Zhang et al., 2024) constructs a draft model by selectively skipping intermediate Transformer blocks and utilizes Bayesian optimization to search for the most effective skip configurations. Building on this, SWIFT (Xia et al., 2025) also employs Bayesian optimization to adaptively select layers to skip during inference, further demonstrating that LLMs exhibit task-specific layer sparsity that can be leveraged for acceleration.

Other training-free methods focus on context-aware adaptation, where the cosine similarity between hidden states plays a pivotal role as a proxy for drafting accuracy. Metel et al. (2024) proposes ASD, a simple layer-skip heuristic that adaptively searches draft configuration based on input context and cosine similarity. Similarly, DEL (Zarch et al., 2025) proposes a dynamic strategy that adjusts both the exit layer and the speculation length during each decoding round. Recently, CLaSp (Chen et al., 2025) employs an in-context layer-skipping strategy that uses a dynamic programming algorithm to optimize the draft configuration by leveraging the cosine similarity of hidden states from previous verification stages. All methods except for LayerSkip and Kangaroo are training-free and can be directly applied to any pre-trained LLM without further modification or auxiliary parameters.

In Table 1, we summarize features of these methods as well as proposed KnapSpec.

## 1.2. Contributions

In this paper, we propose KnapSpec, a framework that reformulates self-speculative decoding (SSD) as a Knapsack Problem to maximize inference throughput. Our contributions can be summarized as follows:

- We extend traditional block-level layer skipping by decoupling Attention and MLP layers and incorporate context length into the draft model selection process. By modeling the hardware-specific latency of attention as a function of sequence length, KnapSpec maintains high speedups even as the context window expands.

- We reformulate draft model selection as a constrained optimization problem. Using a hardware-aware TPT (Tokens-per-Time) metric, KnapSpec balances the draft model's latency and the acceptance rate (Section 3).

- While prior works use cosine similarity as an empirical heuristic, no formal theoretical connection has been established yet. We provide the first rigorous analysis demonstrating how cosine similarity serves as a reliable proxy for the token acceptance rate (Section 4).

- Experiments demonstrate that KnapSpec consistently achieves superior wall-clock speedups compared to state-of-the-art training-free SSD baselines (Section 5).

## 2. Preliminaries

For any vectors $x, y \in \mathbb{R}^d$, we use $\cos(x, y) := \frac{\langle x,y \rangle}{\|x\|_2 \|y\|_2}$ and extend this to a matrix $X, Y \in \mathbb{R}^{n \times d}$ by denoting $\cos(X, Y) = \frac{1}{n} \sum_{i=1}^{n} \cos(X_{i,:}, Y_{i,:})$ where $X_{i,:}$ is the $i$-th row vector in $X$. Throughout this paper, we consider Transformer (Vaswani et al., 2017) architecture for both target and draft models. In particular, a target model is assumed as a Transformer with $L$ layers, where each layer consists of a self-attention module followed by a

multilayer perceptron (MLP)[1]. Let $f$ denote the forward mapping of the target model, which can be expressed as the composition: $f = f_{\text{MLP}}^{(L)} \circ f_{\text{Attn}}^{(L)} \circ \cdots \circ f_{\text{MLP}}^{(1)} \circ f_{\text{Attn}}^{(1)}$ where $f_{\text{MLP}}^{(i)}$ and $f_{\text{Attn}}^{(i)}$ represent the forward functions of the MLP and attention in the $i$-th layer, respectively. For simplicity, we flatten the sequence of layers by defining $f^{(2i-1)} := f_{\text{Attn}}^{(i)}$, $f^{(2i)} := f_{\text{MLP}}^{(i)}$ for $i \in [L]$. Under this scheme, the target model is represented as a composition of $2L$ sequential layers. For any index subset $S \subseteq [2L]$, $f^{(S)}$ denotes the forward mapping of the sub-network formed by composing the layers in $S$ while preserving their original sequential order.

## 2.1. Speculative Decoding

Speculative Decoding (SD) (Leviathan et al., 2023; Chen et al., 2023) accelerates autoregressive decoding by using a fast draft model to generate candidate tokens, which a larger target model verifies in parallel. This approach reduces the generation latency by replacing multiple slow autoregressive calls with parallel verification, significantly lowering overall latency.

Formally, let $\tilde{s}_{t+1:t+\gamma} = (\tilde{s}_{t+1}, \ldots, \tilde{s}_{t+\gamma})$ be $\gamma$ tokens proposed by a draft model $q$. The target model $p$ computes distributions for these candidates in parallel. A token $\tilde{s}_{t+i}$ is accepted if $u \leq \min\left(1, \frac{p(\tilde{s}_{t+i}|s_{<t+i})}{q(\tilde{s}_{t+i}|s_{<t+i})}\right)$ for $u \sim U(0,1)$ in sampling and if $\arg\max_{\tilde{s}_{t+i}} p(\tilde{s}_{t+i}|s_{<t+i}) = \arg\max_{\tilde{s}_{t+i}} q(\tilde{s}_{t+i}|s_{<t+i})$ in greedy decoding.

The efficiency of SD is governed by the acceptance rate as well as the cost of the draft model. The acceptance rate $\alpha \in [0, 1]$ represents the probability that a draft token satisfies the verification criterion. For a fixed draft length $\gamma$, the number of accepted tokens follows a truncated geometric distribution. Hence, the expected number of tokens generated per speculative step is given by

$$\mathbb{E}[\text{Tokens per step}] = \sum_{i=0}^{\gamma} \alpha^i = \frac{1 - \alpha^{\gamma+1}}{1 - \alpha}. \quad (1)$$

## 2.2. Self-Speculative Decoding with Layer Selection

To guarantee the effective benefits of SD, a draft model must operate with low latency while maintaining sufficient alignment with the target model's predictions. This typically requires targeted training or fine-tuning since a draft model with low acceptance rates fails to provide a speedup and instead introduces verification overhead that degrades performance. Prior work commonly introduces a separate draft model (Leviathan et al., 2023), however, this approach is often suboptimal as maintaining two independent sets

of model parameters substantially increases the memory footprint.

Self-Speculative Decoding (SSD) addresses this limitation by leveraging the redundancy of the target model. Rather than introducing an external draft model, SSD constructs the draft as a sub-network of the target model by strategically selecting a subset of its layers.

In the SSD framework, the draft model is represented as $f^{(S)}$ for a properly chosen subset $S$. Increasing the size of $S$ generally improves the approximation quality to the target model, leading to higher acceptance rates of draft tokens, but incurs higher per-token latency. Therefore, the objective of SSD is to find an index set $S$ that optimally balances draft model latency and token acceptance rate, thereby maximizing overall decoding speed. Recent advances in SSD configure sub-networks to serve as draft models and fall into two categories: early-exit and layer-skip strategies.

**Early-Exit SSD.** A line of works such as DEL (Zarch et al., 2025) and LayerSkip (Elhoushi et al., 2024)[2] construct draft models by truncating the target model at some early layer $\ell < L$. DEL (Zarch et al., 2025) proposes a dynamic early-exit strategy by optimizing the Tokens-per-Layer (TPL) metric, which measures the expected number of accepted tokens per loaded layer. Let $\alpha_\ell$ denote the acceptance rate at layer $\ell$ estimated from cached shadow tokens. For an exit layer $\ell$ and speculation length $\gamma$, the TPL objective is defined as:

$$\text{TPL}(\ell, \gamma) := \frac{1 - \alpha_\ell^{\gamma+1}}{1 - \alpha_\ell} \cdot \frac{1}{\gamma\ell + L} \quad (2)$$

where the denominator $\gamma\ell + L$ represents the total number of layers operated including $\gamma$ drafting steps at layer $\ell$ and a single full verification pass. While early-exit methods reduce the search space to a linear prefix $S \in \{[2], [4], \ldots, [2L]\}$ they inherently limit the potential for speedup by forcing the inclusion of all early layers, regardless of their individual utility.

**Layer-Skip SSD.** In contrast, *layer-skip* methods formulate draft construction as a general layer selection problem. This flexibility allows the draft model to have deeper or more informative layers, which are often crucial for SSD performance (Zhang et al., 2024). However, the corresponding search space grows exponentially with depth as $S \subseteq [2L]$, and hence the optimization problem becomes more challenging. To address this, SWIFT (Xia et al., 2025) applies Bayesian optimization over layer subsets based on observations on different combinations of selected layers. Recently, CLaSp (Chen et al., 2025) proposes a layer-skip strategy

---

[1] We include layer normalization to the corresponding Attention or MLP layer.

[2] Despite its name, LayerSkip primarily employs an early-exit strategy rather than skipping intermediate layers.

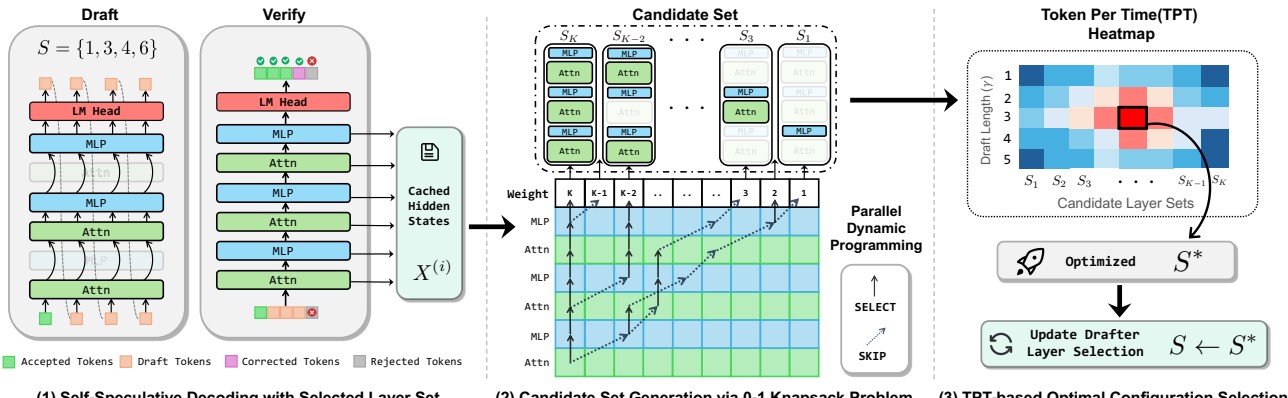

*Figure 1.* Overview of KnapSpec. (1) Self-Speculative Decoding defines a draft model as a sub-network of the target model. (2) Layer selection can be converted to a Knapsack Problem, and we search candidate sets via Dynamic Programming (DP) with various latency budgets (weights). (3) Then, the optimal configuration that maximizes Tokens-per-Time (TPT) is selected and used for the next speculation.

that seeks to maximize the cosine similarity between the draft and target representations under a fixed budget $B$ on the number of layers to draft:

$$\max_{S \subseteq [2L]} \cos(f(x), f^{(S)}(x)) \quad \text{s.t.} \quad |S| = B. \quad (3)$$

Since the problem in Equation (3) is NP-hard, CLaSp employs a dynamic programming (DP) formulation to efficiently obtain a near-optimal layer subset.

## 3. KnapSpec: Layer Selection for SSD as a Knapsack Problem

In this section, we introduce KnapSpec, a novel layer selection framework for Self-Speculative Decoding (SSD). Figure 1 illustrates an overview of KnapSpec. While prior SSD works optimize various proxy metrics, such as Tokens-per-Layer (TPL) or cosine similarity, these metrics often fail to align with actual wall-clock speedups. KnapSpec is designed to bridge this gap by directly aligning the optimization objective with decoding throughput. By decoupling the Attention and MLP layers, we reformulate the selection task as a *0/1 Knapsack Problem*.

### 3.1. Motivation: Asymmetric Layer Latencies

Previous SSD methods often simplify the search space by treating Transformer layers as inseparable units (Zarch et al., 2025; Chen et al., 2025) or assuming uniform latencies across all layers (Xia et al., 2025). However, such static assumptions fail to capture the latency asymmetry inherent in Transformer architectures, which becomes increasingly pronounced as the context length grows. In practice, the latency of an Attention layer grows linearly with the context length $n$, i.e., $t_{\text{Attn}} = \Theta(n)$, while MLP latency remains constant, i.e., $t_{\text{MLP}} = \Theta(1)$.

This divergence implies that a draft configuration optimized for a short context may become inefficient as the computational bottleneck shifts from MLP to Attention during generation. Consequently, the optimal layer selection should not be a global or static configuration. Instead, it must be adaptively determined based on the current context length. KnapSpec addresses this by incorporating pre-profiled, length-dependent latency as dynamic weights in our optimization framework. By doing so, KnapSpec ensures that the drafting strategy is always tailored to the actual latency of the layers at the given context, maximizing throughput throughout the entire long-context inference process.

### 3.2. Tokens-per-Time (TPT) Metric

To address the limitations of existing efficiency proxies that lack direct alignment with wall-clock speed, we introduce the Tokens-per-Time (TPT) metric. TPT generalizes the Tokens-per-Layer (TPL) metric in Equation (2) by explicitly accounting for the distinct, length-dependent latencies of Attention and MLP layers. Specifically, TPT represents the expected number of generated tokens per unit of wall-clock time:

$$\text{TPT}(S, \gamma) = \frac{1 - \alpha_S^{\gamma+1}}{1 - \alpha_S} \cdot \frac{1}{\gamma \, t_{\text{Draft}}(S) + t_{\text{Target}}} \quad (4)$$

where $\alpha_S$ is the expected acceptance rate for the layer set $S$ and $\gamma$ is the draft length. The right term represents the total latency required for a single speculation step, where $t_{\text{Draft}}(S)$ is the drafting time and $t_{\text{Attn}}$ is the verification time of the target model. These latencies are calculated as:

$$t_{\text{Draft}}(S) = n_{\text{Attn}}(S) \cdot t_{\text{Attn}} + n_{\text{MLP}}(S) \cdot t_{\text{MLP}},$$
$$t_{\text{Target}} = L(t_{\text{Attn}} + t_{\text{MLP}})$$

where $n_{\text{Attn}}(S)$ and $n_{\text{MLP}}(S)$ denote the number of Attention and MLP layers in $S$, respectively. We determine the co-

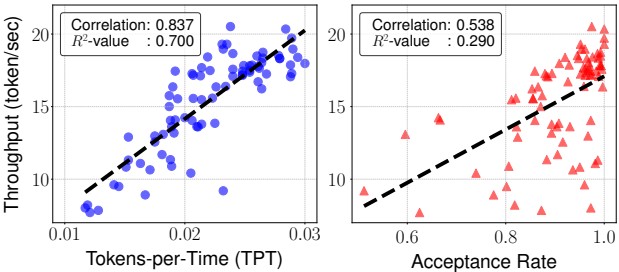

*Figure 2.* Pearson Correlation Coefficients (PCC) and $R^2$-value of TPT and acceptance rate against throughput. Best TPT shows a much closer correlation with actual performance.

efficients $t_{\text{Attn}}$ and $t_{\text{MLP}}$ by measuring hardware-dependent latency during a one-time preprocessing step across varying context length and hardware specification. See Section C in the Appendix for further details.

By replacing the number of layers with the actual execution time, TPT provides a more precise objective for maximizing throughput. In scenarios where $t_{\text{Attn}}$ and $t_{\text{MLP}}$ are assumed to be equal, as in previous works (Zarch et al., 2025; Chen et al., 2025), optimizing TPT simplifies to the optimization of TPL. As illustrated in Figure 2, TPT shows a much higher correlation with practical throughput (i.e., number of generated tokens per time) compared to the estimated accepted rate, demonstrating that TPL is a superior metric to represent actual speed, with further details in Section 5.2.

### 3.3. Problem Formulation as a 0/1 Knapsack Problem

Our goal is to find the optimal layer set $S^*$ and draft length $\gamma^*$ that maximize the TPT metric:

$$S^*, \gamma^* = \underset{S \subseteq [2L], \gamma \in [D]}{\operatorname{argmax}} \ \text{TPT}(S, \gamma) \tag{5}$$

where $D$ is the maximum draft length. Since Equation (5) involves an exponential search space of size $2^{2L}D$, direct optimization is intractable.

To solve this efficiently, we decompose the problem into two stages by leveraging the fact that for a fixed drafting latency $t_{\text{Draft}}(S)$, TPT is strictly increasing with respect to the acceptance rate $\alpha_S$. In the first stage (Section 3.4), we reformulate the layer selection task as a *0/1 Knapsack Problem* to identify optimal sub-networks that maximize the expected acceptance rate across various latency budgets. Specifically, each Attention and MLP layer is defined as an *item*, where its latency serves as the *weight* and its cosine similarity to the target model representations serves as the *value*. As calculating $\alpha_S$ directly is costly, we utilize cosine similarity as a reliable proxy to represent the value, a choice we formally justify in Section 4. By solving this problem for a range of feasible budgets, we obtain a candidate set $\mathcal{A} = \{S_k\}$, where each $S_k$ represents the layer set that maximizes the acceptance rate for a given latency budget $k$.

---

**Algorithm 1** Dynamic Programming for Layer Selection

1: **Input:** hidden state matrices $\{X^{(i)}\}_{i \in [2L]}$, decoder mappings $\{f^{(i)}\}_{i \in [2L]}$, layer weights $(w_{\text{Attn}}, w_{\text{MLP}})$
2: $K \leftarrow (w_{\text{Attn}} + w_{\text{MLP}}) \cdot L$
3: $g \leftarrow \text{zeros}(2L + 1, K + 1, r, d)$
4: $g[0, 0] \leftarrow X^{(0)}$          // Forward Search
5: **for** $i = 1$ to $2L$ **do**
6:      $w \leftarrow w_{\text{Attn}}$ **if** $i \bmod 2 = 1$ **else** $w_{\text{MLP}}$
7:      **for** $j = 0$ to $K$ **in parallel do**
8:          **if** $g[i - 1, j] \neq \mathbf{0}$ **then**
9:              $h_{\text{e}} \leftarrow f^{(i)}(g[i - 1, j])$
10:              $\sigma_{\text{e}} \leftarrow \cos(h_{\text{e}}, X^{(i)})$
11:          **if** $\sigma_{\text{e}} > \max(\tau, \cos(g[i, j], X^{(i)}))$ **then**
12:              $g[i, j] \leftarrow h_{\text{e}}$
13:          $h_{\text{s}} \leftarrow g[i - 1, j]$
14:          $\sigma_{\text{s}} \leftarrow \cos(h_{\text{s}}, X^{(i)})$
15:          **if** $\sigma_{\text{s}} > \max(\tau, \cos(g[i, j + w], X^{(i)}))$ **then**
16:              $g[i, j + w] \leftarrow h_{\text{s}}$
17: $\mathcal{A} \leftarrow \emptyset$          // TPT Maximization
18: **for** $j = 0$ to $K$ **do**
19:      **if** $g[2L, j] \neq \mathbf{0}$ **then**
20:          $S \leftarrow \text{Backtrack}(g, 2L, j)$
21:          $\widehat{\alpha}_S \leftarrow \text{AccRate}(g[2L, j], X^{(2L)})$    ▷ Eq.(9)
22:          $\mathcal{A} \leftarrow \mathcal{A} \cup \{S\}$
23: $S^*, \gamma^* \leftarrow \operatorname{argmax}_{(S, \gamma) \in \mathcal{A} \times [D]} \frac{\mathbb{E}[\text{Tokens}(\widehat{\alpha}_S, \gamma)]}{\text{Cost}(S, \gamma)}$  ▷ Eq.(4)
24: **return** $S^*$

---

In the second stage (Section 3.5), we perform a final optimization over this reduced search space. Since $\mathcal{A}$ contains the selections that maximize the proxy for $\alpha_S$ for every possible drafting cost, the global optimum $(S^*, \gamma^*)$ is expected to reside within $\mathcal{A} \times [D]$. We calculate the actual TPT for each candidate pair and select the one that maximizes TPT.

### 3.4. Efficient Optimization via Dynamic Programming

In the first stage, we use a parallel Dynamic Programming (DP) algorithm that processes the representations obtained from generation. The detailed algorithm of the optimization process is provided in Algorithm 1.

**Integer Weight Normalization.** To utilize DP, we first transform continuous latencies into discrete integer weights by approximating $t_{\text{Attn}}(n) \approx w_{\text{Attn}}\Delta$ and $t_{\text{MLP}} \approx w_{\text{MLP}}\Delta$, where $\Delta = \min(t_{\text{Attn}}(n), t_{\text{MLP}})$ serves as the base unit and the weights are normalized through rounding: $w_{\text{Attn}} = \lfloor t_{\text{Attn}}(n)/\Delta \rceil$ and $w_{\text{MLP}} = \lfloor t_{\text{MLP}}(n)/\Delta \rceil$. Then, the latency $t_{\text{Draft}}(S)$ becomes $(n_{\text{Attn}}(S)w_{\text{Attn}} + n_{\text{MLP}}(S)w_{\text{MLP}})\Delta$.

For a fixed integer budget $k$, we define the corresponding

Knapsack problem to find a layer set $S$ as

$$\begin{aligned} \underset{S \subseteq [2L]}{\text{argmax }} \alpha_S &\approx \underset{S \subseteq [2L]}{\text{argmax }} \cos(f(X), f^{(S)}(X)) \\ \text{s.t.} \quad & n_{\texttt{Attn}}(S)w_{\texttt{Attn}} + n_{\texttt{MLP}}(S)w_{\texttt{MLP}} = k, \end{aligned} \quad (6)$$

where $X \in \mathbb{R}^{r \times d}$ is the matrix of concatenated representations for the $r$ tokens generated from recent $m = 5$ speculation steps. This formulation allows us to partition the space of possible layer sets into discrete latency bins, enabling systematic exploration of the trade-off between the expected acceptance rate and execution latency.

**Recurrence Relation in DP.** Dynamic Programming (DP) provides an efficient, near-optimal solution for Equation (6). We define $g[i, j] \in \mathbb{R}^{r \times d}$ as the optimal hidden representations achieved at layer $i$ with a skipped weight $j$. For each step, we consider two candidate states representing the *execution* and *skipping* of the current layer:

$$h_{\texttt{e}} = f^{(i)}(g[i-1, j]), \quad h_{\texttt{s}} = g[i-1, j - w_i], \quad (7)$$

where $w_i \in \{w_{\texttt{Attn}}, w_{\texttt{MLP}}\}$. With the base condition $g[0, 0] = X^{(0)}$, the DP recurrence selects the candidate that have higher cosine similarity to the reference representation $X^{(i)} = f^{(i)}(X^{(i-1)}) \in \mathbb{R}^{r \times d}$:

$$g[i, j] = \begin{cases} h_e & \text{if } \sigma_{\texttt{e}} \geq \sigma_{\texttt{s}} \\ h_s & \text{otherwise} \end{cases} \quad (8)$$

where the similarity scores $\sigma_{\texttt{e}} = \cos(h_e, X^{(i)})$ and $\sigma_{\texttt{s}} = \cos(h_s, X^{(i)})$. Consequently, each entry $g[2L, j]$ in the terminal row encodes the optimal sequence of layer selections for a skip budget $j$, collectively forming the candidate pool for our final TPT optimization.

**Single-pass Efficiency.** A key advantage of this formulation is that a single execution of the DP algorithm with a maximum budget $K = L(w_{\texttt{Attn}} + w_{\texttt{MLP}})$ simultaneously yields optimal solutions for all budget partitions $k \leq K$. By storing the maximum attainable similarity for each discrete latency bin, we avoid redundant computations when searching for the global optimum. Furthermore, since the forward functions $f^{(i)}$ and cosine operations are independent across the budget dimension $j$, they are computed in parallel using batch processing.

**Efficiency via Pruning.** To further minimize overhead, we employ two pruning criteria to discard paths where: (i) the cosine similarity with the reference state falls below $\tau = 0.5$, or (ii) the total skipped weight exceeds $K/2$. These criteria are based on our empirical observation that states failing these conditions are highly likely to be sub-optimal due to excessive accuracy degradation. Removing such unpromising paths reduces search overhead in time and memory without sacrificing the final selection quality. We provide further results on various $\tau$ in Section 5.2.

## 3.5. TPT Maximization via Grid Search

The second stage identifies the optimal layer set $S^*$ that maximizes TPT from the candidate set $\mathcal{A} = \{S_k\}_{k \in [K]}$ derived via backtracking from the terminal DP table entries $g[2L, K - k]$, as detailed in Algorithm 1.

For each candidate $S \in \mathcal{A}$, we estimate the expected acceptance rate $\widehat{\alpha}_S$ by evaluating its prediction accuracy in the recent history $X$:

$$\widehat{\alpha}_S = \frac{1}{r} \sum_{i=1}^{r} \mathbb{I}(\widehat{y}^{(S)}(X_{i,:}) = y(X_{i,:})) \quad (9)$$

where $\widehat{y}^{(S)}(X_{i,:})$ is the token predicted by $i$-th embedding $X_{i,:}$ and configuration $S$, and $y(X_{i,:})$ is the next ground-truth token from the target model. We then perform a grid search over $\mathcal{A} \times [D]$ to select the optimal pair $(S^*, \gamma^*)$ that maximizes the TPT objective ($D = 10$ in our experiment).

**Dynamic Drafting Exit.** Even though we obtain $\gamma^*$, we instead use an adaptive $\gamma$ based on real-time token confidence during inference to achieve additional efficiency gains. Following prior work (Chen et al., 2025; Zarch et al., 2025), we terminate the drafting process if the top-1 probability of the draft token falls below $\tau_{\text{conf}} = 0.7$, based on the observation that the acceptance rate of draft tokens is highly correlated with this confidence score (Li et al., 2024). This prevents wasted computation on low-confidence tokens that are likely to be rejected, further maximizing throughput.

## 3.6. Discussion on Runtime and Memory Complexity

Our two-stage process effectively reduces the exponential search space of Equation (5) with size $2^{2L}D$ to a tractable set. By retaining only $D$ candidates for each $S \in \mathcal{A}$, the search space reduces to $O(nD)$ as $|\mathcal{A}| = K$ scales linearly with the context length $O(n)$. A naïve implementation of the DP algorithm would require $O(KL)$ forward mapping calls. Since each layer takes $O(n)$ time, the time complexity is $O(n^2 L)$ and the memory overhead becomes $O(nL)$. We optimize this using a parallel DP algorithm. By leveraging batch processing of hidden representations, we reduce runtime and memory complexity to $O(nL)$ and $O(L)$, respectively. This ensures that the search overhead remains the same complexity as the standard autoregressive decoding, while transforming the exponential dependency on $L$ into a linear one.

# 4. Cosine Similarity in Layer Selection SSD

In this section, we rigorously investigate the relationship between the cosine similarity of hidden embeddings and the speculative acceptance rate. Although CLaSp (Chen et al., 2025) empirically utilized the cosine similarity between the target output $f(x)$ and the draft output $f^{(S)}(x)$ as a proxy

*Table 2.* Performance comparison of reasoning tasks (long-context generation) and summarization tasks (long-context input). We use Qwen3 models for reasoning (left) and Llama3 models for summarization (right) with various scales. We report Tokens-per-Time (TPT), wall-clock speedup, and the average acceptance rate $\alpha$ for speculative decoding methods.

| Methods | Reasoning Tasks (Long-context Generation) | | | | | | | | | Summarization Tasks (Long-context Input) | | | | | | | | |
| | AIME24 | | | AIME25 | | | MMLU-Pro | | | GovReport | | | PG19 | | | BookSum | | |
| | TPT | Speedup | $\alpha$ | TPT | Speedup | $\alpha$ | TPT | Speedup | $\alpha$ | TPT | Speedup | $\alpha$ | TPT | Speedup | $\alpha$ | TPT | Speedup | $\alpha$ |
|---|---|---|---|---|---|---|---|---|---|---|---|---|---|---|---|---|---|---|
| | Qwen3-32B ($L = 64$ layers) | | | | | | | | | Llama3.1-70B ($L = 80$ layers) | | | | | | | | |
| AR | 19.65 | 1.00× | – | 19.72 | 1.00× | – | 23.15 | 1.00× | – | 6.75 | 1.00× | – | 6.90 | 1.00× | – | 4.76 | 1.00× | – |
| DEL | 19.69 | 0.85× | 0% | 19.95 | 0.88× | 2.8% | 23.20 | 0.87× | 0% | 6.62 | 0.87× | 0.1% | 6.99 | 0.87× | 1.3% | 4.84 | 0.87× | 0.0% |
| SWIFT | 20.38 | 1.23× | 84.6% | 20.73 | 1.22× | 80.4% | 21.75 | 0.90× | 46.8% | 8.62 | 1.33× | 91.6% | 6.12 | 0.98× | 61.9% | 3.76 | 0.84× | 49.3% |
| CLaSp | 21.65 | 1.30× | 96.1% | 21.30 | 1.29× | 95.1% | 23.90 | 1.09× | 92.0% | 6.82 | 1.22× | 93.4% | 6.52 | 1.10× | 87.8% | 4.57 | 1.01× | 81.3% |
| KnapSpec | 31.06 | **1.43×** | 93.5% | 32.00 | **1.42×** | 93.1% | 34.62 | **1.20×** | 85.1% | 13.00 | **1.47×** | 94.4% | 10.67 | **1.21×** | 89.5% | 6.99 | **1.14×** | 84.0% |
| | Qwen3-14B ($L = 40$ layers) | | | | | | | | | Llama3.1-8B ($L = 32$ layers) | | | | | | | | |
| AR | 22.45 | 1.00× | - | 23.17 | 1.00× | - | 24.06 | 1.00× | - | 15.71 | 1.00× | - | 16.11 | 1.00× | - | 10.77 | 1.00× | - |
| DEL | 22.48 | 0.83× | 0.1% | 23.25 | 0.83× | 0% | 24.61 | 0.85× | 17.2% | 15.72 | 0.74× | 0.1% | 16.12 | 0.71× | 0.1% | 11.49 | 0.74× | 0.1% |
| SWIFT | 22.38 | 1.14× | 84.4% | 21.86 | 1.20× | 85.4% | 13.67 | 0.74× | 53.6% | 13.72 | 1.05× | 62.1% | 12.43 | 0.83× | 47.7% | 8.25 | 0.72× | 31.6% |
| CLaSp | 21.14 | 1.19× | 92.3% | 21.42 | 1.20× | 92.5% | 15.82 | 1.08× | 87.0% | 14.15 | 1.11× | 91.7% | 13.94 | 1.10× | 91.4% | 8.79 | 1.05× | 85.3% |
| KnapSpec | 30.53 | **1.30×** | 86.1% | 31.06 | **1.27×** | 88.9% | 35.51 | **1.16×** | 88.0% | 25.00 | **1.28×** | 97.0% | 23.91 | **1.26×** | 94.6% | 15.48 | **1.18×** | 91.5% |
| | Qwen3-8B ($L = 36$ layers) | | | | | | | | | Llama3.2-3B ($L = 28$ layers) | | | | | | | | |
| AR | 25.15 | 1.00× | – | 28.66 | 1.00 | – | 24.97 | 1.00× | – | 22.58 | 1.00 | – | 23.54 | 1.00 | – | 15.71 | 1.00 | – |
| DEL | 22.96 | 0.82× | 0% | 28.71 | 0.83× | 0% | 27.91 | 0.88× | 23.9% | 23.72 | 0.84× | 36.0% | 25.62 | 0.68× | 15.5% | 16.37 | 0.75× | 20.1% |
| SWIFT | 20.60 | 1.12× | 75.0% | 21.65 | 0.90× | 71.0% | 20.32 | 1.12× | 93.0% | 18.45 | 1.08× | 64.0% | 18.82 | 0.83× | 53.6% | 11.09 | 0.89× | 42.1% |
| CLaSp | 21.84 | 1.14× | 92.4% | 23.94 | 1.15× | 90.5% | 22.75 | 1.16× | 90.8% | 18.91 | 1.08× | 85.7% | 21.10 | 1.08× | 91.0% | 12.22 | 1.06× | 82.3% |
| KnapSpec | 33.47 | **1.28×** | 89.8% | 36.81 | **1.26×** | 88.5% | 40.01 | **1.33×** | 91.2% | 36.47 | **1.24×** | 95.2% | 34.04 | **1.16×** | 92.1% | 23.48 | **1.18×** | 87.0% |
| | Qwen3-4B ($L = 36$ layers) | | | | | | | | | Llama3.2-1B ($L = 16$ layers) | | | | | | | | |
| AR | 34.78 | 1.00× | – | 36.85 | 1.00× | – | 30.93 | 1.00× | – | 46.56 | 1.00× | – | 46.68 | 1.00× | – | 30.43 | 1.00× | – |
| DEL | 40.77 | 0.98× | 44.9% | 40.60 | 0.97× | 22.3% | 38.36 | 0.99× | 66.1% | 49.31 | 0.89× | 55.6% | 49.98 | 0.77× | 32.9% | 32.97 | 0.76× | 31.4% |
| SWIFT | 27.54 | 0.88× | 80.3% | 30.23 | 1.11× | 89.0% | 23.74 | 1.00× | 7.50% | 31.32 | 1.03× | 83.8% | 31.34 | 0.91× | 71.5% | 21.31 | 0.96× | 69.5% |
| CLaSp | 29.68 | 1.15× | 90.9% | 32.26 | 1.16× | 91.2% | 25.68 | 1.15× | 89.9% | 33.75 | 1.02× | 83.6% | 33.61 | 1.01× | 81.7% | 23.42 | 1.04× | 82.8% |
| KnapSpec | 46.20 | **1.36×** | 87.9% | 49.96 | **1.35×** | 89.3% | 45.25 | **1.29×** | 90.2% | 58.79 | **1.11×** | 96.1% | 58.17 | **1.06×** | 91.7% | 38.23 | **1.13×** | 91.2% |

to identify draft models, the theoretical justification remains largely intuitive. We bridge this gap by demonstrating that sufficiently high cosine similarity is a sufficient condition for draft token acceptance, providing a formal justification for its use as an optimization objective.

Suppose that embeddings in the LM head are given by $w_1, ..., w_V \in \mathbb{R}^d$ where $V$ is the size of the vocabulary. In greedy decoding, the predicted token $y(x)$ for the last hidden embedding $x$ becomes $\text{argmax}_{j \in [V]} \langle w_j, x \rangle$. We identify a condition on the cosine similarity between the target embedding $x$ and a draft embedding $x'$ that guarantees identical next token selection.

**Lemma 4.1.** *Given embeddings $w_1, ..., w_V \in \mathbb{R}^d$ and a fixed vector $x \in \mathbb{R}^d$, let $i^* := \text{argmax}_i \langle w_i, x \rangle$ and define the margin $\xi(x) := \langle w_{i^*}, x \rangle - \max_{j \neq i^*} \langle w_j, x \rangle$. For any $x' \in \mathbb{R}^d$ such that $\|x'\|_2 = \|x\|_2$, if*

$$\cos(x, x') \geq 1 - \frac{\xi(x)^2}{2\|x\|_2^2 \max_{j \neq i^*} \|w_{i^*} - w_j\|_2^2} \quad (10)$$

*then $\text{argmax}_{i \in [V]} \langle w_i, x \rangle = \text{argmax}_{i \in [V]} \langle w_i, x' \rangle$.*

The proof is provided in Section A. Lemma 4.1 demonstrates that maximizing the cosine similarity of draft hidden states is a mathematically sound strategy for minimizing divergence from the target model's greedy predictions. The assumption $\|x\|_2 \approx \|x'\|_2$ is empirically supported by modern architectures (e.g., RMSNorm) that project hidden states

onto a hypersphere of nearly constant radius.

As a result, maximizing cosine similarity under a latency budget (i.e., Equation (3)) serves as a reliable surrogate for the greedy acceptance rate $\alpha_S$. While not identical, Lemma 4.1 establishes cosine similarity as a rigorous proxy for ensuring the draft model is close to the target model: $\text{argmax}_S \alpha_S \approx \text{argmax}_S \cos(f(X), f^{(S)}(X))$. This allows KnapSpec framework to maximize throughput across varying context lengths while maintaining provable guarantees on the consistency of the generated tokens.

## 5. Experiments in Long-context Scenarios

We evaluate KnapSpec against state-of-the-art training-free SSD methods: SWIFT (Xia et al., 2025), DEL (Zarch et al., 2025), and CLaSp (Chen et al., 2025). Our evaluation spans diverse long-context scenarios: (1) long-context generation using reasoning tasks from AIME24/25 (Zhang & Math-AI, 2024) and MMLU-Pro (Wang et al., 2024) datasets with Qwen3 (Yang et al., 2025) models, and (2) long-context input processing using summarization tasks from GovReport (Huang et al., 2021), PG19 (Rae et al., 2020), and BookSum (Kryściński et al., 2022) datasets with Llama3 (Grattafiori et al., 2024) models.

For reasoning tasks, the maximum generation length is set to 32K tokens for AIME24/25, and 4K tokens for MMLU-Pro. Summarization tasks involve an average input length

*Table 3.* Performance comparison under nucleus sampling with a temperature $T = 0.7$, and top-$k$ and top-$p$ thresholds set to $k = 50$ and $p = 0.95$, respectively. We use Qwen3-8B model for reasoning (left) and Llama3.1-8B model for summarization (right). We report Tokens-per-Time (TPT), wall-clock speedup, and the average acceptance rate $\alpha$ for speculative decoding methods.

| Methods | Reasoning Tasks (Qwen3-8B) | | | | | | | | | Summarization Tasks (Llama3.1-8B) | | | | | | | | |
|---|---|---|---|---|---|---|---|---|---|---|---|---|---|---|---|---|---|---|
| | AIME24 | | | AIME25 | | | MMLU-Pro | | | GovReport | | | PG19 | | | BookSum | | |
| | TPT | Speedup | $\alpha$ | TPT | Speedup | $\alpha$ | TPT | Speedup | $\alpha$ | TPT | Speedup | $\alpha$ | TPT | Speedup | $\alpha$ | TPT | Speedup | $\alpha$ |
| AR | 31.87 | 1.00× | – | 32.58 | 1.00× | – | 33.52 | 1.00× | – | 19.71 | 1.00× | – | 20.64 | 1.00× | – | 14.02 | 1.00× | – |
| DEL | 31.29 | 0.86× | 0.0% | 37.89 | 0.82× | 0.0% | 25.58 | 0.83× | 22.2% | 20.01 | 0.63× | 0.0% | 20.64 | 0.67× | 0.1% | 13.91 | 0.69× | 0.0% |
| SWIFT | 24.46 | 1.03× | 73.3% | 27.15 | 1.02× | 78.1% | 20.72 | 1.11× | 89.3% | 16.28 | 0.87× | 48.2% | 14.38 | 0.52× | 24.7% | 12.78 | 0.47× | 22.4% |
| CLaSp | 24.08 | 1.06× | 83.1% | 27.62 | 1.07× | 91.6% | 24.68 | 1.09× | 91.4% | 16.98 | 1.07× | 86.0% | 15.83 | 1.02× | 82.3% | 9.03 | 1.02× | 78.0% |
| KnapSpec | 47.66 | **1.13×** | 79.3% | 49.96 | **1.10×** | 87.9% | 49.89 | **1.19×** | 89.6% | 32.16 | **1.23×** | 93.6% | 28.59 | **1.07×** | 83.8% | 17.56 | **1.08×** | 83.9% |

of up to 16K tokens. We set the optimization interval to $T = 64$ steps for models with fewer than 10B parameters and $T = 128$ for larger variants.

## 5.1. Main Performance Results

As shown in Table 2, KnapSpec consistently outperforms all baselines in both Tokens-per-Time (TPT) values and wall-clock speedup across the 1B to 70B parameter range. While existing methods struggle to maintain efficiency as the context window expands, KnapSpec remains robust. For instance, on GovReport with Llama3.1-70B, KnapSpec achieves a 1.47× speedup, significantly exceeding SWIFT (1.33×) and CLaSp (1.22×). Notably, DEL fails to provide acceleration in this setting, as it is designed for fine-tuned models to be robust to early-exit, such as LayerSkip, hence not guaranteed to work effectively for general base models.

This performance gain demonstrates that by decoupling Attention and MLP layers and optimizing for hardware-aware latency, KnapSpec bypasses the throughput bottlenecks inherent in traditional block-level strategies. Furthermore, KnapSpec consistently achieves the highest TPT across all datasets. Notably, while competing methods occasionally yield higher acceptance rates, their actual throughput remains lower than ours. This confirms that the acceptance rate is an insufficient proxy for wall-clock efficiency. By directly optimizing TPT via our Knapsack optimization, KnapSpec identifies configurations that perfectly balance predictive accuracy with execution cost.

We further evaluate the performance of KnapSpec under stochastic sampling settings beyond greedy decoding. Specifically, we employ nucleus sampling (Holtzman et al., 2020) with a temperature of $T = 0.7$, and top-$k$ and top-$p$ thresholds set to $k = 50$ and $p = 0.95$, respectively. As summarized in Table 3, KnapSpec consistently maintains its execution efficiency and wall-clock speedup even when tokens are generated probabilistically. This confirms that directly optimizing for length-dependent hardware latencies via our parallel dynamic programming knapsack selection provides consistent robustness across both deterministic and non-deterministic decoding strategies.

## 5.2. Ablation Studies

**TPT vs. Acceptance Rate for Practical Speedup.** We investigate whether TPT or acceptance rate is a better predictor of practical speedup. Using Llama-3.1-8B on Gov-Report (12K–13K tokens), we compute the Pearson correlation between these metrics and final throughput. As shown in Figure 2, TPT shows a significantly higher correlation (0.837) with throughput than with the acceptance rate (0.538). While a higher acceptance rate is generally desirable as it indicates better model alignment, it often ignores the computational overhead of the drafting process or architectural bottlenecks in local speculative execution. In contrast, TPT inherently balances the expected token against the wall-clock time required for generation, making it a more predictive indicator for optimizing speculative decoding.

**Adaptive Layer Selection Ratio.** We analyze the layer selection behaviors of KnapSpec compared to SWIFT (Xia et al., 2025) using the Qwen3-8B on the AIME24. For SWIFT, we apply Bayesian optimization independently at each context length $n$ (20 separate optimization runs each) to identify the optimal configuration. As the latency of the Attention layer grows linearly with $n$ while the MLP latency remains constant, Attention becomes the dominant bottleneck in long-context scenarios. As shown in Figure 3, KnapSpec naturally increases the ratio of skipped Attention layers as $n$ grows, demonstrating its ability to adaptively optimize to increasing costs. In contrast, SWIFT's selection remains relatively static due to its lack of an explicit latency-aware objective. This highlights the need for a search mechanism that can identify a dynamic, length-aware configuration to maintain high decoding speed as $n$ increases.

**Cosine Similarity Pruning Threshold.** We evaluate the impact of the similarity threshold $\tau$ in Section 3.4 using Llama-3.1-8B. We prune candidate states where the cosine similarity with the reference falls below $\tau$ to reduce the search space and memory usage. In Figure 4, $\tau = 0.5$ maintains a speedup of 1.26×, matching the speedup without pruning while reducing memory consumption by 31%. This indicates that pruning based on the cosine similarity

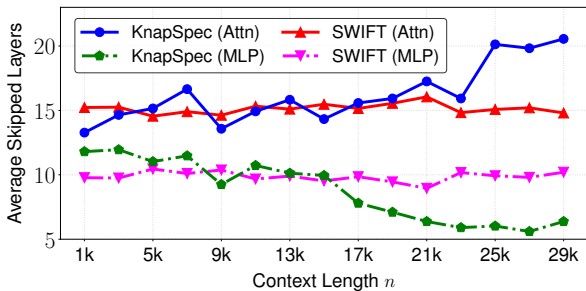

*Figure 3.* Comparison of skipped layers across context lengths up to 32k. KnapSpec chooses more Attention layers to skip than MLP as the length $n$ increases, while SWIFT chooses uniform layers.

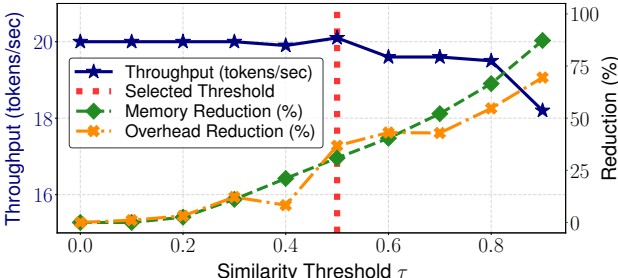

*Figure 4.* Throughput, memory and optimization overhead reduction (%) with various similarity thresholds $\tau$. We select $\tau = 0.5$ to achieve the best balance between speed and memory.

effectively filters redundant branches without sacrificing performance. However, exceeding this threshold causes a decline in speedup as valid candidates are prematurely discarded. Thus, we select $\tau = 0.5$ to preserve maximum speedup while improving search efficiency.

## 6. Conclusion

We introduce KnapSpec, a training-free self-speculative decoding framework that formulates the layer selection task as a knapsack problem. By decoupling Attention and MLP layers and accounting for hardware-aware, length-dependent latencies, KnapSpec adaptively identifies optimal draft configurations to maximize decoding throughput. Furthermore, we provide a rigorous theoretical foundation for using cosine similarity as a reliable proxy for the expected acceptance rate. Extensive evaluations across various LLMs and datasets demonstrate superior speedups, particularly in long-context scenarios. Our work offers a practical and efficient solution for accelerating next-generation generative AI without additional training or intensive overhead.

## Acknowledgments

This work was supported by the National Research Foundation of Korea (NRF) grant funded by the Korea government (MSIT) (No. RS-2024-00406715 and No. RS-2025-23523958) and BK21 FOUR(Connected AI Education & Research Program for Industry and Society Innovation, KAIST EE, No. 4120200113769). It was also supported in part by U.S. NSF IIS-2225942. Any opinions, findings, conclusions, or recommendations expressed in this material are those of the authors and do not necessarily reflect the views of these organizations.

## Impact Statement

This paper presents work whose goal is to advance the field of Machine Learning. There are many potential societal consequences of our work, none of which we feel must be specifically highlighted here.

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

## A. Proof of Theorems

**Lemma 4.1.** *Given embeddings $w_1, ..., w_V \in \mathbb{R}^d$ and a fixed vector $x \in \mathbb{R}^d$, let $i^* := \arg\max_i \langle w_i, x \rangle$ and define the margin $\xi(x) := \langle w_{i^*}, x \rangle - \max_{j \neq i^*} \langle w_j, x \rangle$. For any $x' \in \mathbb{R}^d$ such that $\|x'\|_2 = \|x\|_2$, if*

$$\cos(x, x') \geq 1 - \frac{\xi(x)^2}{2 \|x\|_2^2 \max_{j \neq i^*} \|w_{i^*} - w_j\|_2^2} \tag{10}$$

*then $\arg\max_{i \in [V]} \langle w_i, x \rangle = \arg\max_{i \in [V]} \langle w_i, x' \rangle$.*

*Proof.* For any $j \neq i^*$, we observe that

$$\begin{aligned}
\langle w_{i^*} - w_j, x' \rangle &= \langle w_{i^*} - w_j, x \rangle + \langle w_{i^*} - w_j, x - x' \rangle \\
&\geq \xi(x) + \langle w_{i^*} - w_j, x - x' \rangle \\
&\geq \xi(x) - \|w_{i^*} - w_j\|_2 \|x - x'\|_2
\end{aligned} \tag{11}$$

where the first inequality comes from the definition of $\xi(x)$ and the second one is from the Cauchy-Schwarz inequality. Equation (10) is equivalent to

$$\|x - x'\|_2 \leq \frac{\xi(x)}{\max_{j \neq i^*} \|w_{i^*} - w_j\|_2}. \tag{12}$$

Combining Equation (11) and (12) yields $\langle w_{i^*} - w_j, x' \rangle \geq 0$, equivalently, this holds $\arg\max_i \langle w_i, x' \rangle = i^*$. $\qquad\square$

## B. Additional Experiments

### B.1. Optimization Interval

We investigate the impact of the optimization interval (in steps) on both end-to-end throughput and optimization overhead. The optimization interval determines how frequently the algorithm re-profiles and updates the layer selection strategy. A smaller interval allows for more fine-grained adaptation to dynamic context changes but incurs higher computational overhead, whereas a larger interval reduces overhead but may lead to stale configurations.

Figure 5 illustrates the trade-off between throughput and optimization overhead. As the interval increases from 4 to 64 steps, optimization overhead decreases significantly (40.9% to 7.5%). Consequently, throughput peaks at 18.1 tok/s with an interval of 64. However, beyond 64 steps (e.g., at 128 and 256), throughput degrades despite minimal overhead. The skip configuration becomes outdated too quickly, reducing the acceptance rate and negating the benefits of lower optimization cost. Thus, an interval of 64 steps provides the best trade-off between generation speed and adaptive responsiveness.

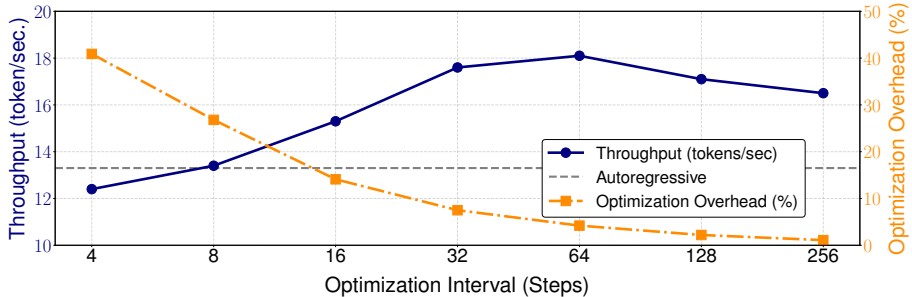

*Figure 5.* The trade-off between throughput (blue) and optimization overhead (orange) as a function of the optimization interval. Increasing the interval reduces overhead but eventually harms throughput due to stale layer configurations. The optimal interval is found at 64 steps, achieving the highest throughput of 18.1 tokens/second.

### B.2. Distribution of Skipped Layers

We analyze the layer-wise skip distribution of the Qwen3-8B model($L = 36$) and Llama-3.1-8B($L = 32$) model using our proposed method. Figure 6 illustrates the probability of skipping each layer, aggregated over the evaluation samples. The layers are indexed from 1 to $2L$, alternating between Attention and MLP layers.

As shown in Figure 6, skip probabilities are non-uniform and model-specific. Certain layers exhibit near-zero skip probabilities, suggesting their essential roles (e.g., 10-th layer in Qwen; 28–33-th layers in Llama). In contrast, other layers are skipped more frequently than their neighbors, indicating higher structural redundancy (e.g., 21-st layer in Qwen; 47-th layer in Llama).

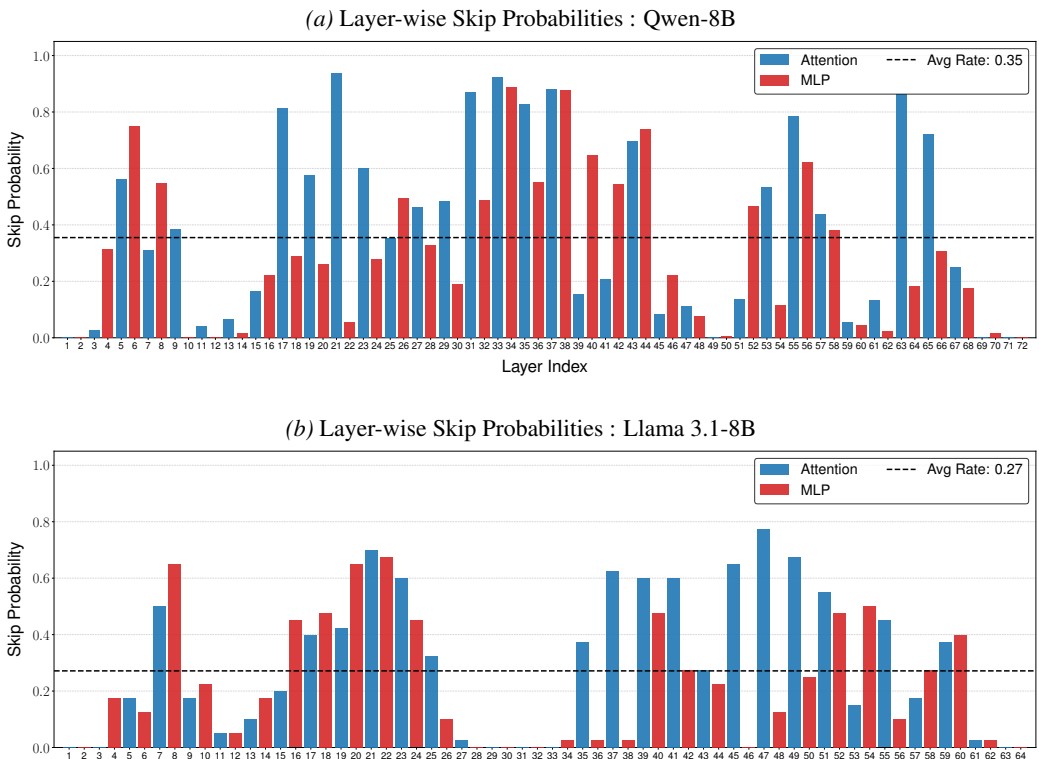

*Figure 6.* Layer-wise skip probabilities aggregated over evaluation samples. The non-uniform distribution demonstrates the model-specific redundancies. While certain layers are frequently skipped to reduce computational cost, some layers consistently show low skip probabilities.

## C. Hardware-Aware Latency Formulation and Robustness

To formalize the length-dependent costs in Section 3.2, we model the latencies of MLP and Attention layers as functions of the context length $n$: $t_{\text{MLP}} = c_1$ and $t_{\text{Attn}}(n) = c_2 \cdot n + c_3$, where $c_1, c_2, c_3$ are hardware- and model-specific empirical coefficients. While MLP latency ($c_1$) is invariant to context length due to static operations, Attention latency ($c_2 \cdot n + c_3$) scales linearly with $n$, accurately capturing system bottlenecks like KV cache access and memory bandwidth in long-context scenarios. We determine these coefficients via least squares linear regression from a one-time offline profiling sweep.

| Hardware | Model | $c_1$ | $c_2$ ($\times 10^{-4}$) | $c_3$ | **Speedup** |
|---|---|---|---|---|---|
| RTX 6000 Pro | Llama-3.1-8B | 0.279 | 0.735 | 0.279 | 1.27× |
| | Llama-3.2-3B | 0.161 | 0.720 | 0.255 | 1.25× |
| RTX A6000 | Llama-3.1-8B | 0.520 | 0.942 | 0.216 | 1.28× |
| | Llama-3.2-3B | 0.248 | 0.855 | 0.186 | 1.24× |
| RTX 4090 | Llama-3.1-8B | 0.386 | 0.766 | 0.167 | *O.O.M.* |
| | Llama-3.2-3B | 0.180 | 0.724 | 0.140 | 1.26× |

*Table 4.* Multi-Hardware Profiling and KnapSpec's Speedup on GovReport.

As summarized in Table 4, while the fitted coefficients vary across diverse GPU architectures and model scales, Knap-

Spec observes consistent speedups (i.e., $\sim 1.25\times$). By plugging these analytically computed weights into the dynamic programming search on the fly, the framework adaptively identifies the TPT-optimal subnetwork at any given token position, confirming the practical validity and robustness of our formulation across diverse system settings.

## D. Algorithm Description

To provide a more granular view of the optimization process introduced in Section 3, we decompose the procedure into two specialized stages. Algorithm 1 details the forward dynamic programming used to populate the similarity table, while Algorithm 2 describes the systematic backtracking and grid search used to identify the optimal configuration.

---

**Algorithm 1** Dynamic Programming for Candidate Search

---

1: **Input:** hidden state matrices $\{X^{(i)}\}_{i\in[2L]}$, decoder mappings $\{f^{(i)}\}_{i\in[2L]}$, layer weights $(w_{\text{Attn}}, w_{\text{MLP}})$, number of layers $L$, hidden size $d$, number of cached tokens $r$
2: $K \leftarrow (w_{\text{Attn}} + w_{\text{MLP}}) \cdot L$         // Initialization
3: $g \leftarrow \text{zeros}(2L + 1, K + 1, r, d)$
4: $g[0,0] \leftarrow X^{(0)}$         // Forward Search (DP)
5: **for** $i = 1$ to $2L$ **do**
6:      $g[i,0] \leftarrow X^{(i)}$
7:      $w \leftarrow w_{\text{Attn}}$ **if** $i \bmod 2 = 1$ **else** $w_{\text{MLP}}$
8:      **for** $j = 0$ to $K$ **in parallel do**
9:          **if** $g[i-1, j] \neq \mathbf{0}$ **then**
10:             $h_{\text{e}} \leftarrow f^{(i)}(g[i-1, j])$
11:          **if** $\cos(h_{\text{e}}, X^{(i)}) > \cos(g[i,j], X^{(i)})$ **then**
12:             $g[i,j] \leftarrow h_{\text{e}}$         // Execute
13:          $h_{\text{s}} \leftarrow g[i-1, j]$
14:          **if** $\cos(h_{\text{s}}, X^{(i)}) > \cos(g[i,j+w], X^{(i)})$ **then**
15:             $g[i, j+w] \leftarrow h_{\text{s}}$         // Skip
16: **return** $g$

---

---

**Algorithm 2** Layer Set Selection Based on TPT Maximization

---

1: **Input:** DP table $g$, final hidden state matrix $X^{(2L)}$, module weight $(w_{\texttt{MLP}}, w_{\texttt{Attn}})$, maximum draft length $D$
2: $S^* \leftarrow \emptyset, \text{TPT}^* \leftarrow 0$          // BackTracking
3: **for** $k = 0$ to $K$ **do**
4:     $h \leftarrow g[2L, k]$
5:     **if** $h = \mathbf{0}$ **then continue**
6:     $S, i, j \leftarrow \emptyset, 2L, k$
7:     **while** $i > 0$ and $j > 0$ **do**
8:        $w \leftarrow w_{\texttt{Attn}}$ **if** $i \bmod 2 = 1$ **else** $w_{\texttt{MLP}}$
9:        **if** $g[i, j] = g[i-1, j-w]$ **then**
10:          $j \leftarrow j - w$
11:        **else**
12:          $S \leftarrow S \cup \{i-1\}$
13:        $i \leftarrow i - 1$
                                              // Acceptance Rate Estimation
14:     $N_{\texttt{match}} \leftarrow 0$                     ▷ Equation (9)
15:     **for** $i = 1$ to $r$ **do**
16:        **if** $\arg\max_{v \in [V]} \langle w_v, h_{i,:} \rangle = \arg\max_{v \in [V]} \langle w_v, X_{i,:}^{(2L)} \rangle$ **then**
17:          $N_{\texttt{match}} \leftarrow N_{\texttt{match}} + 1$
18:     $\widehat{\alpha}_S \leftarrow N_{\texttt{match}}/r$                     // Grid Search
19:     **for** $\gamma = 1$ to $D$ **do**
20:        $\text{TPT} \leftarrow \frac{1-\widehat{\alpha}_S^{\gamma+1}}{1-\widehat{\alpha}_S} \cdot \frac{1}{\gamma\, t_{\texttt{Draft}}(S) + t_{\texttt{Target}}}$       ▷ Equation (4)
21:        **if** $\text{TPT} > \text{TPT}^*$ **then**
22:          $S^* \leftarrow S, \quad \text{TPT}^* \leftarrow \text{TPT},$
23: **return** $S^*$

---

