# OpenReview forum: "KnapSpec: Self-Speculative Decoding via Adaptive Layer Selection as a Knapsack Problem"
_ICML.cc/2026/Conference — ICML 2026 regular_

### Official Review · Reviewer_XDww · 2026-03-09

**Soundness:** 3
**Presentation:** 2
**Significance:** 3
**Originality:** 3
**Overall Recommendation:** 4
**Confidence:** 4

**Summary:**

This paper proposes KnapSpec, a training-free framework for accelerating large language model inference using self-speculative decoding (SSD). The method constructs a draft model by selecting a subset of layers from the target model and formulates the layer-selection problem as a 0/1 knapsack optimization that balances drafting latency and token acceptance rate. By decoupling attention and MLP layers and incorporating context-length–dependent latency into a Tokens-per-Time (TPT) objective, KnapSpec adaptively selects efficient draft configurations during decoding. The authors also provide a theoretical justification for using cosine similarity between hidden states as a proxy for token acceptance. Experiments on Qwen3 and Llama3 models show that KnapSpec consistently improves decoding throughput, achieving up to 1.47× speedup over existing training-free SSD methods.

**Compliance With Llm Reviewing Policy:**

Affirmed.

**Final Justification:**

This paper proposes an interesting optimization for self-speculative decoding and provides solid experiments. However, I hold a conservative view on the future of self-speculative decoding. Therefore, I gave a weak accept rating.

**Key Questions For Authors:**

1\. How robust is cosine similarity as a proxy for selecting draft layer sets across different tokens and execution modes?
The proposed method relies on cosine similarity to rank candidate layer sets and approximate the acceptance rate. However, cosine similarity appears to be evaluated when the verification stage runs the full model, while during drafting only a subset of layers is activated. It is unclear whether the ranking of layer sets remains consistent under partial-layer execution. In addition, since the layer configuration is updated using a small set of recent tokens, the ranking may vary across tokens. The authors could analyze this by reporting per-token ranking correlations (e.g., Kendall tau) and comparing the rankings obtained under full-layer and partial-layer execution.

2\. How does the proposed method compare with speculative decoding approaches designed for long-context inference, such as methods that construct draft models using sparse attention or other long-context acceleration techniques?

3\. What are the main factors limiting the speedup of the proposed method?

**Note:** The first question is particularly important, as it relates directly to a core assumption underlying the proposed optimization objective. The second question can be addressed with a limited additional experiment (e.g., one model and one dataset) to provide a clearer empirical comparison.

**Limitations:**

No. The paper does not sufficiently discuss practical limitations. In particular, the reported speedup is relatively modest, and many modern LLMs already incorporate pretraining-time acceleration techniques (e.g., multi-token prediction), which raises questions about the practical usefulness of the proposed method.

**Strengths And Weaknesses:**

The paper studies an important problem in large language model inference: improving the efficiency of self-speculative decoding (SSD) without requiring additional training or auxiliary draft models. The proposed framework reformulates the draft model selection problem as a knapsack optimization, which directly considers the trade-off between draft latency and token acceptance rate.

A notable design choice is the decoupling of Attention and MLP layers when constructing the draft model. Instead of treating each Transformer block as an indivisible unit, the method models their latency separately and dynamically decides whether to skip them. This design is particularly meaningful for long-context inference, where the cost of attention grows with sequence length. By explicitly modeling these asymmetric costs and solving the selection problem via a dynamic programming approach, the paper provides a principled way to adapt the draft configuration based on runtime characteristics.

Despite the interesting formulation, several aspects of the paper limit the overall impact of the proposed approach.

First, the overall acceleration remains relatively limited. Compared with the autoregressive baseline, the reported speedups range from roughly 1.06× to 1.43×, which is significantly smaller than the improvements typically observed in mainstream speculative decoding approaches. While it is understandable that many speculative decoding methods rely on auxiliary draft models or additional training (e.g., mtp/eagle heads), the gains reported here appear relatively modest. This raises questions about the practical benefits of self-speculative decoding approaches in general.

Second, since the method specifically targets long-context inference, the experimental evaluation should include comparisons with speculative decoding methods that leverage sparse attention to construct draft models[1][2]. Such approaches may provide competitive or complementary benefits in long-sequence scenarios.

Third, the use of cosine similarity as a proxy for token acceptance raises a potential methodological concern. In the current formulation, cosine similarity is evaluated when the verification stage runs the full model, meaning that all layers are activated. However, during the drafting stage only a subset of layers is executed. It is unclear whether the relative contribution of different layers to cosine similarity remains consistent when the network is partially evaluated. In other words, the influence of each layer on representation similarity may change when intermediate layers are skipped, which could affect the reliability of the optimization objective.

[1] Sadhukhan R, Chen J, Chen Z, et al. Magicdec: Breaking the latency-throughput tradeoff for long context generation with speculative decoding[J]. arXiv preprint arXiv:2408.11049, 2024.

[2] Sun H, Chen Z, Yang X, et al. Triforce: Lossless acceleration of long sequence generation with hierarchical speculative decoding[J]. arXiv preprint arXiv:2404.11912, 2024.

---

> ### Author Rebuttal · Authors · 2026-03-31
>
> Thank you for your thoughtful review and for recognizing the importance of the problem we address, as well as the practical value of the KnapSpec framework. We appreciate your positive assessment of our principled approach to decoupling Attention and MLP layers for efficient long-context inference. Below, we address your specific comments to further clarify the validity of our formulation and its comparative advantages over existing speculative decoding methods.
>
> ### [W1 & Q3: Speedup Constraints and Practicality]
> We focus on self-speculative decoding (SSD) because of its high practicality. This approach requires no auxiliary models and no additional training, and zero extra VRAM. KnapSpec provides a "plug-and-play" acceleration that is immediately applicable to any LLM, with no offline preprocessing or layer-selection optimization. While mainstream training-based methods (e.g., Medusa, EAGLE) can offer higher absolute speedups, they require substantial deployment overhead and memory costs. These costs are often prohibitive for rapidly updated models or memory-constrained environments.
>
> By design, SSD relies on the target LLM's inherent structural redundancy without an auxiliary model. Thus, the maximum speedup is naturally bounded by the minimum number of layers required for accurate drafting. Moreover, our on-the-fly DP search introduces a minor computational overhead (approx. 5–15% of inference time). Despite these constraints, KnapSpec consistently outperforms existing training-free SSD baselines (e.g., CLaSp, SWIFT), especially in long-context scenarios where hardware-aware efficiency becomes the primary bottleneck.
>
> ---
>
> ### [W3 & Q1: Validity of Cosine Similarity Proxy]
> We respectfully clarify that KnapSpec does not rank a small set of pre-defined candidates.  Because the search space of possible subnetworks is exponential ($O(2^{2L})$), computing per-token ranking correlations (e.g., Kendall tau) is computationally infeasible. Instead, our Dynamic Programming (DP) algorithm is designed to efficiently identify and directly construct the optimal layer combination.
>
> Regarding concerns about partial execution, our DP formulation explicitly accounts for the intermediate state shifts caused by skipped layers at each DP step. By accumulating these representation shifts during the DP process, our optimization objective reflects the true behavior of the partial-layer execution.
>
> To ensure stability, we identify the optimal layer set using a sequence of recently generated tokens, which smooths out local variations and generalizes well to upcoming tokens. This process dynamically adapts to both the changing hidden states and the increasing context length. Such continuous adaptation ensures that the draft configuration remains highly effective throughout the generation, leading to consistently high empirical acceptance rates and practical speedups.
>
>
> ---
>
>
> ### [W2 & Q2: Comparison with Long-context SD Methods]
> KnapSpec is orthogonal to speculative decoding methods that leverage sparse attention (e.g., MagicDec[1], TriForce[2]). While those methods focus on compressing the context dimension (width) by reducing KV cache access, KnapSpec optimizes the model dimension (depth) through adaptive layer selection. Because these two approaches address different computational bottlenecks, they are highly complementary. KnapSpec’s hardware-aware layer selection can be applied on top of sparse-attention-based models to further enhance throughput. We will update our Related Work to discuss these methods and highlight their integration as a promising direction in our Future Work.
>
> [1] Sadhukhan R, Chen J, Chen Z, et al. Magicdec: Breaking the latency-throughput tradeoff for long context generation with speculative decoding[J]. arXiv preprint arXiv:2408.11049, 2024.
>
> [2] Sun H, Chen Z, Yang X, et al. Triforce: Lossless acceleration of long sequence generation with hierarchical speculative decoding[J]. arXiv preprint arXiv:2404.11912, 2024.

---

> > ### Author Rebuttal · Reviewer_XDww · 2026-04-02
> >
> > I appreciate the clarification from the authors. However, because MTP has been integrated into mainstream pre-training paradigm, I am conservative regarding the overall future impact of self-speculative decoding. I will maintain my score and but raise my confidence accordingly.

---

> > > ### Author Response · Authors · 2026-04-08
> > >
> > > Thank you again for your time and carefully considering our rebuttal.
> > > It appears you attempted to increase the confidence score via the comments section, but the system has not yet been updated. We would appreciate it if you could look into this.
> > >
> > > Regarding the future impact of SSD alongside multi-token prediction (MTP), we respectfully clarify that our approach is orthogonal to MTP. While MTP modifies the pretraining objective, KnapSpec operates as a training-free, plug-and-play inference accelerator. As such, it can be directly applied to existing models, including those with MTP, without retraining or architectural changes. In fact, combining MTP with KnapSpec could further improve overall decoding efficiency.
> > >
> > > In addition, we would like to respectfully follow up on your Question 1, which you highlighted as particularly important, as it directly concerns the core assumption underlying our method. As clarified, our DP formulation explicitly accounts for intermediate state shifts during partial-layer execution, and we ensure stability across tokens by dynamically adapting the layer configuration based on recently generated sequences.
> > >
> > > If your concerns are fully resolved, we respectfully ask if you might consider a positive adjustment to your score. We deeply appreciate your time and support!

---

### Official Review · Reviewer_PxjG · 2026-03-11

**Soundness:** 3
**Presentation:** 3
**Significance:** 2
**Originality:** 3
**Overall Recommendation:** 4
**Confidence:** 3

**Summary:**

This paper proposes KnapSpec, a training-free self-speculative decoding (SSD) framework that adaptively selects a subnetwork of the target LLM for drafting by formulating layer selection as a 0/1 knapsack problem.  Its key idea is to decouple Attention and MLP layers and to use context-length–aware, hardware-profiled latencies as weights, while using cosine similarity as a value proxy for acceptance rate.

By combining this quality proxy with hardware-aware latency modeling, KnapSpec searches for better draft configurations for long-context inference. Experiments on Llama3 and Qwen3 models show up to 1.47× end-to-end speedup over recent training-free SSD baselines on long-context generation and summarization tasks.

**Compliance With Llm Reviewing Policy:**

Affirmed.

**Final Justification:**

While some limitations remain, particularly in the breadth of validation and the fact that some of the added support appears in the rebuttal rather than the main paper, taking both the paper and the rebuttal into account, I raise my final recommendation to Weak Accept.

**Key Questions For Authors:**

1.	The current theory and experiments are largely confined to greedy decoding. It is important to clarify whether KnapSpec remains effective under stochastic sampling, where cosine similarity may be a weaker proxy for acceptance and practical speedup.
2.	The method relies heavily on a hardware-aware latency model and dynamic-programming search. A clearer analysis of how robust this formulation is under very long contexts and different hardware settings would be important for judging its practical validity.
3.	The paper would benefit from a clearer positioning within the broader speculative decoding landscape, especially relative to draft-model-based SD methods. This is important for understanding in which scenarios KnapSpec is the right design choice and what trade-offs define its practical niche.

**Limitations:**

No.

**Strengths And Weaknesses:**

**Strengths**

1.  The paper is well motivated by a practical limitation of prior self-speculative decoding methods, namely that acceptance rate does not reliably reflect real wall-clock throughput, especially in long-context settings where Attention latency becomes dominant.

2.   The proposed method is intuitive, as it decouples Attention and MLP layers and formulates draft subnetwork selection as a knapsack problem with context-length-aware latency costs, yielding a more fine-grained and hardware-aware alternative to block-level, length-agnostic designs.

3.  The approach is training-free, easy to deploy, and the experiments show consistent throughput and speedup improvements across multiple model sizes and long-context tasks, supporting its real-world utility.

**Weaknesses**

1.	The quality–speed trade-off is not consistently favorable across all reasoning tasks. In Table 2, KnapSpec achieves higher speedup than CLaSP in some settings, but this sometimes comes with a noticeable drop in accuracy, for example on AIME24 and MMLU-Pro with Qwen3-32B. This makes the practical benefit less clear in accuracy-sensitive reasoning scenarios.
2.	The theoretical analysis and main experiments are largely restricted to greedy decoding. Since many practical generation settings rely on stochastic sampling, it remains unclear whether cosine similarity is still a reliable proxy for draft quality and whether the reported speedup gains would transfer to non-greedy inference.
3.	The method depends on a simplified hardware-aware latency model and a dynamic-programming search procedure, but the robustness of this formulation under broader system settings is not fully established. The current analysis does not sufficiently address how well the latency model captures real bottlenecks such as KV-cache access and memory bandwidth limits, especially at very long context lengths.

---

> ### Author Rebuttal · Authors · 2026-03-31
>
> Thank you for the constructive feedback. We have addressed all concerns with additional experiments and corrected the initial notation confusion by clarifying that KnapSpec is strictly lossless with zero accuracy drop by design. Given these results, we kindly ask you to consider a positive re-evaluation of our work.
>
> ### [W1: Metric Clarification ("Acc." in Table 2 means **Acceptance Rate** not Accuracy)]
> The term “Acc.” in Table 2 refers to the **Acceptance Rate** ($\alpha$) not **Accuracy**. This is the average percentage of drafted tokens accepted by the target model, and not the task performance accuracy. We will update this terminology in the revised manuscript to ensure clarity and prevent any further misunderstanding.
>
> We emphasize that KnapSpec, like any other speculative decoding methods, is a strictly lossless acceleration technique. Since the target LLM validates all drafted tokens, the final output of KnapSpec is mathematically identical to that of the original model under greedy decoding. Therefore, there is **no degradation in task-level accuracy** or generation quality by design compared to the autoregressive baseline or other speculative methods like CLaSp.
>
> ---
>
> ### [W2 & Q1: Analysis under Stochastic Sampling]
> The choice of cosine similarity is rooted in maximizing feature-level alignment between the draft subnetwork and the target model. In stochastic sampling, the acceptance rate is determined by the overlap between the draft and target probability distributions, which can be directly quantified by the Total Variation Distance. By minimizing the distance in the feature space, we inherently minimize the divergence between these output distributions.
>
> Furthermore, we show the expected acceptance rate $\alpha$ under stochastic sampling can be lower-bounded as a function of the cosine similarity (i.e., Sampling version of Lemma 4.1) below. Informally, we get $\alpha \geq 1 - \frac{{\max_{j} \|\|w_j\|\|_2 \|\|x\|\|_2}}{\beta} \sqrt{2(1 - \cos(x, x'))}$, where $\beta$ is the temperature parameter. We will include the full statement and proof in the final version.
>
> KnapSpec remains effective in stochastic sampling scenarios. For the comprehensive results, please refer to **Table 1 and Table 2** provided in our response to Reviewer CTXx. Our experiments using standard sampling hyperparameters ($T=0.7, \text{top-}k=50, \text{top-}p=0.95$) on a single NVIDIA RTX 6000 PRO GPU confirm that KnapSpec consistently achieves higher speedups than other SSD baselines under stochastic sampling. Although absolute speedups are lower than in greedy decoding due to the increased variance, these stable throughput improvements prove our feature-level alignment with the cosine similarity proxy remains highly effective for practical acceleration beyond greedy settings.
>
> ---
>
> ### [W3 & Q2: Hardware-Aware Latency Formulation and Robustness]
> Our model ($t_{\mathtt{MLP}}=c_1, t_{\mathtt{Attn}}(n)=c_2 \cdot n + c_3$) uses direct wall-clock profiling to capture system bottlenecks like KV-cache access and memory bandwidth. While MLP latency ($c_1$) is invariant to context length, Attention latency ($c_2 \cdot n + c_3$) scales linearly with $n$, accurately reflecting overhead in long-context scenarios.
>
> Table 3 shows that while coefficients vary by hardware (i.e., GPU) and model, KnapSpec observe consistent speedups (i.e., ~1.25x) by adapting DP search to select the TPT-optimal subnetwork. This confirms the practical validity and robustness of our formulation across diverse system settings.
>
> **Table 3: Multi-Hardware Profiling and KnapSpec’s Speedup (GovReport)**
>
> | Hardware | Model | $c_1$ | $c_2 (\times 10^{-4})$ | $c_3$ | Speedup |
> | :--- | :--- | :---: | :---: | :---: | :---: |
> | RTX 6000 Pro | Llama-3.1-8B | 0.279 | 0.735 | 0.279 | 1.27x |
> | | Llama-3.2-3B | 0.161 | 0.720 | 0.255 | 1.25x |
> | RTX A6000 | Llama-3.1-8B | 0.520 | 0.942 | 0.216 | 1.28x |
> | | Llama-3.2-3B | 0.248 | 0.855 | 0.186 | 1.24x |
> | RTX 4090 | Llama-3.1-8B | 0.386 | 0.766 | 0.167 | O.O.M. |
> | | Llama-3.2-3B | 0.180 | 0.724 | 0.140 | 1.26x |
>
> ---
>
> ### [Q3: Trade-offs and Practical Scenarios]
> The fundamental trade-off between draft-model-based speculative decoding and KnapSpec lies in Absolute Speedup vs. Deployment Overhead. While training-based methods can offer higher acceleration, they require significant offline training costs and a persistent VRAM footprint for an auxiliary model. In contrast, KnapSpec offers zero training cost and zero extra memory footprint, making its practical niche ideal for memory-constrained environments where an auxiliary draft model cannot fit alongside the target LLM, or for customized or rapidly evolving models where maintaining and training a separate draft model for every update is unfeasible. By utilizing the target model’s inherent structural redundancy, KnapSpec provides a robust, "plug-and-play" acceleration for real-world deployments where resource efficiency and deployment agility are the primary constraints.

---

> > ### Author Rebuttal · Reviewer_PxjG · 2026-04-03
> >
> > The rebuttal largely addressed my concern about stochastic sampling by providing both additional theoretical discussion and empirical evidence beyond the greedy setting. I will raise the score to 4 (weak accept).

---

> > > ### Author Response · Authors · 2026-04-08
> > >
> > > Thank you for your highly constructive feedback and for raising your score. We are happy that our additional analyses successfully addressed your concerns and appreciate your time and effort in helping us improve the quality and rigor of our work. We will integrate additional results on sampling version into the final version of our work.

---

### Official Review · Reviewer_CTXx · 2026-03-26

**Soundness:** 2
**Presentation:** 4
**Significance:** 3
**Originality:** 3
**Overall Recommendation:** 4
**Confidence:** 3

**Summary:**

This paper studies how to reduce the inference cost of large language models through self-speculative decoding. In long-context settings, the attention latency becomes the dominant. The authors propose a training-free framework KnapSpec that formulates draft model selection in SSD as the standard knapsack problem. The key idea is to
decouple the Attention and MLP layers, and model their hardware-specific latencies as functions of context length. KnapSpec can dynamically select the optimal configuration by parallel dynamic programming followed by a grid search.

They propose the metric Tokens-per-Time (TPT), which is more directly aligned with wall-clock throughput. Also, the paper proves that the cosine similarity between hidden states is a sound proxy for the token acceptance rate.

**Compliance With Llm Reviewing Policy:**

Affirmed.

**Final Justification:**

In general, the paper proposes a training-free, plug-and-play inference accelerator, which I believe will be of interest to the ICML audience and worthy of further research. During the rebuttal, the authors provided detailed explanation and supplemented their work with additional experimental results in sampling scenarios. Therefore, I'm inclined to support acceptance.

**Key Questions For Authors:**

* Lemma 4.1 requires $\cos(x,x')$ is lower bounded, does it provide any guidance for choosing $\tau^*$? You mention that the equal norm assumption is empirically supported by modern architectures, could you provide some empirical evidence in your own settings?
* The integer weight normalization part is not clear to me. Why a unified $\Delta$ for both latencies is reasonable for analysis and algorithm, instead of using different base unit? Please clarify the detail. For example, $t_{Draft}$ includes a term $n_{Attn}(S)t_{Attn}(n)$ where $t_{Attn}(n)$ is the verification time rather than drafting time, why not use $n_{Attn}(S)t_{Draft}$?
* The paper studies under greedy decoding setting. What about the result in sampling setting?

**Limitations:**

* While the paper has given a theoretical explanation that why cosine similarity is a successful proxy, many other components of the method remain primarily empirical, such as the parameter selection for pruning, dynamic drafting exit and speculation steps, which might be of theoretical interest.

**Strengths And Weaknesses:**

## Strengths:
* The paper is well-structured and easy to follow, and the content is well-explained, especially the Introduction part and the problem formulation (Sec 3.3).
* The insight to optimizes a hardware-aware TPT is the key strength of the paper, which makes the method better aligned with actual wall-clock throghput.
* Knapsack-style algorithm is elegant, which provides a principled way to handle the exponentially large search space, and parallel DP makes the search practically tractable.
* The experiment result are strong and showcase the superiority over other SSD methods over various long-context scenarios, supporting the TPT metric.

## Weakness:
* The theoretical analysis is somewhat limited and simple. There remains a gap between the analysis of cosine similarity as a proxy and empirical use.
* Many results/components are still empirical or heuristic, such as the integer weight normalization, the pruning threshold, and the dynamic drafting exit mechanism, whose rationale could be presented more clearly.
* See Questions below.

---

> ### Author Rebuttal · Authors · 2026-03-31
>
> Thank you for your careful review and for recognizing the high quality of our presentation and the elegant, principled nature of our hardware-aware TPT optimization. We appreciate your positive assessment of the insights behind KnapSpec and its strong experimental results in long-context scenarios. Below, we address your points regarding the cosine similarity proxy, latency modeling, and performance under sampling.
>
> ### [W1 & Q1: Cosine Similarity Lower Bound and Equal-Norm Assumption]
> Lemma 4.1 provides a mathematical justification for using cosine similarity as a principled proxy for token agreement, rather than prescribing a universal theoretical $\tau^*$. In practice, the pruning threshold $\tau$ is a configurable parameter chosen to reflect the actual distribution of cosine similarities.
>
> Statistical analysis on Llama-3.1-8B (GovReport) using 128 hidden embeddings shows that 92.97% of $\cos(x, x')$ values naturally exceed 0.5, explaining why the speedup remains stable for $\tau \leq 0.5$. Setting $\tau=0.5$ prunes the search space and reduces memory footprint by 31% while preserving the throughput gains.
>
> From the same experimental setting, we also provide empirical evidence for the equal-norm assumption: for the norm ratio $\|\|x'\|\|_2 / \|\|x\|\|_2$, we observed a mean of 1.003, a std of 0.019, a median of 1.000, and quartiles [0.991, 1.015]. These results confirm that hidden representations are tightly matched in norm space.
>
> ---
>
> ### [W2 & Q2: Integer Weight Normalization and Latency Modeling]
> KnapSpec maximizes Tokens-per-Time (TPT) by solving a resource-constrained optimization problem during decoding. We use integer weight normalization with a shared base unit $\Delta$ to map continuous Attention and MLP latencies onto a 1D integer axis. While multi-dimensional DP is theoretically possible, it significantly increases search complexity and memory overhead. By using a single $\Delta$, we ensure the search remains computationally tractable during decoding while directly aligning with the wall-clock time budget.
>
> Regarding latency, we define $t_{\mathtt{Attn}}(n)$ and $t_{\mathtt{MLP}}$ as profiled unit latencies. We use the same values for both drafting and verification because the per-layer latency for $k$ tokens is nearly identical to that for a single token during drafting, since verification of $k$ tokens runs in parallel. This consistency allows KnapSpec to accurately predict costs: $t_{\mathtt{Draft}} = n_{\mathtt{Attn}} \cdot t_{\mathtt{Attn}}(n) + n_{\mathtt{MLP}} \cdot t_{\mathtt{MLP}}$ and $t_{\mathtt{Target}} = L(t_{\mathtt{Attn}}(n) + t_{\mathtt{MLP}})$. This unified modeling enables KnapSpec to reliably select the TPT-optimal subnetwork.
>
> ---
>
> ### [Q3: Results in Sampling]
> KnapSpec remains effective in sampling scenarios. Our experiments using standard sampling hyperparameters ($T=0.7, \text{top-}k=50, \text{top-}p=0.95$) on a single NVIDIA RTX 6000 Pro GPU confirm that KnapSpec consistently achieves higher speedups than other SSD baselines under sampling.
>
> **Table 1: Reasoning Results (Qwen3-8B)**
>
> | | | AIME24 | |  | AIME25 | | | MMLU-pro | |
> | :--- | ---: | ---: | ---: | ---: | ---: | ---: | ---: | ---: | ---: |
> | **Method** | TPT | Speedup | $\alpha$ | TPT | Speedup | $\alpha$ | TPT | Speedup | $\alpha$ |
> | AR | 31.87 | 1.00$\times$ | - | 32.58 | 1.00$\times$ | - | 33.52 | 1.00$\times$ | - |
> | DEL | 31.29 | 0.86$\times$ | 0.0 | 37.89 | 0.82$\times$ | 0.0 | 25.58 | 0.83$\times$ | 22.2 |
> | SWIFT | 24.46 | 1.03$\times$ | 73.36 | 27.15 | 1.02$\times$ | 78.1 | 20.72 | 1.11$\times$ | 89.3 |
> | CLaSp | 24.08 | 1.06$\times$ | 83.1 | 27.62 | 1.07$\times$ | 91.6 | 24.68 | 1.09$\times$ | 91.4 |
> | KnapSpec | 47.66 | **1.13$\times$** | 79.3 | 49.96 | **1.10$\times$** | 87.9 | 49.89 | **1.19$\times$** | 89.6 |
>
> **Table 2: Summarization Results (Llama3.1-8B)**
>
> | | | GovReport | | | PG19 | | | BookSum | |
> | :--- | ---: | ---: | ---: | ---: | ---: | ---: | ---: | ---: | ---: |
> | **Method** | TPT | Speedup | $\alpha$ | TPT | Speedup | $\alpha$ | TPT | Speedup | $\alpha$ |
> | AR | 19.71 | 1.00$\times$ | - | 20.64 | 1.00$\times$ | - | 14.02 | 1.00$\times$ | - |
> | DEL | 20.01 | 0.63$\times$ | 0.0 | 20.64 | 0.67$\times$ | 0.1 | 13.91 | 0.69$\times$ | 0.0 |
> | SWIFT | 16.28 | 0.87$\times$ | 48.2 | 14.38 | 0.52$\times$ | 24.7 | 12.78 | 0.47$\times$ | 22.4 |
> | CLaSp | 16.98 | 1.07$\times$ | 86.0 | 15.83 | 1.02$\times$ | 82.3 | 9.03 | 1.02$\times$ | 78.0 |
> | KnapSpec | 32.16 | **1.23$\times$** | 93.6 | 28.59 | **1.07$\times$** | 83.8 | 17.56 | **1.08$\times$** | 83.9 |
>
> Note that $\alpha$ in the tables represent the acceptance rate of draft tokens.

---

> > ### Author Rebuttal · Reviewer_CTXx · 2026-04-03
> >
> > I appreciate the authors’ clarification. I have no further questions and will maintain my original score.

---

> > > ### Author Response · Authors · 2026-04-08
> > >
> > > Thank you for reviewing our response and confirming that all your questions have been adequately addressed. We deeply appreciate your constructive engagement.
> > > Since you noted having no further questions and selected “Fully resolved,” we would be deeply grateful if you could re-evaluate your score in light of this resolution. Thank you again for your time and valuable suggestions.

---

### Decision · Program_Chairs · 2026-04-30

**Decision:**

Accept (regular)

**Comment:**

This paper introduces KnapSpec, a training‑free self‑speculative decoding framework that formulates draft subnetwork selection as a hardware‑aware knapsack problem, explicitly optimizing tokens‑per‑time (TPT) under long‑context inference. Reviewers broadly agree that the formulation is well motivated and practically useful, particularly the decoupling of Attention and MLP layers and the dynamic programming solution that adapts to context length and hardware latency. The plug‑and‑play nature, absence of auxiliary models, and solid empirical speedups (up to ~1.47×) make the method well-suited for deployment.
Concerns remain that several components—latency modeling, pruning thresholds, dynamic exit rules—are partly heuristic, and that absolute speedups are more modest than training‑based speculative decoding methods.
Overall, this is a solid and interesting contribution with a realistic scope. I recommend Weak Accept.